# Comparison of revised EWGSOP2 criteria of sarcopenia in patients with cancer using different parameters of muscle mass

**Ana Paula Trussardi Fayh** [1,2,3] *, **Iasmin Matias de Sousa** [1,2]

**1** Postgraduate Program in Health Sciences, Health Sciences Center, Federal University of Rio Grande do Norte, Natal, RN, Brazil, **2** Postgraduate Program in Nutrition, Health Sciences Center, Federal University of Rio Grande do Norte, Natal, RN, Brazil, **3** Postgraduate Program in Physical Education, Health Sciences Center, Federal University of Rio Grande do Norte, Natal, RN, Brazil

* apfayh@yahoo.com.br

## Abstract

Calf circumference (CC) has been established as a marker of muscle mass (MM) with good performance for predicting survival in individuals with cancer. The study aims to determine the prevalence of sarcopenia according to the European Working Group on Sarcopenia in Older People 2 (EWGSOP2) criteria and to evaluate the accuracy of sarcopenia using low CC relative to MM assessment by computed tomography (CT) at third lumbar vertebra level (L3) as a reference. Cross-sectional study with cancer patients aged $\geq$ 60 years. Data included socio-demographic, clinical and anthropometric variables. MM was assessed by CC and by CT images at the L3. Sarcopenia was diagnosed according to the EWGSOP2 criteria: a) low handgrip strength (HGS) + reduced MM evaluated by CT; and b) low HGS + low CC. Pearson's correlation, accuracy, sensitivity, specificity, positive predictive and negative predictive value were analyzed. A total of 108 patients were evaluated, age of 70.6 ± 7.4 years (mean ± standard deviation). The prevalence of sarcopenia was of 24.1% (low MM) and 25.9% (low CC). The Kappa test showed a substantial agreement (K = 0.704), 81% sensitivity, and 92% specificity. Although the EWGSOP2 advises that we should use CC measures in the algorithm for sarcopenia when no other MM diagnostic methods are available, the findings allow the use of CC instead of MM by CT in cancer patients.

## Introduction

People worldwide are living longer. According to the World Health Organization (WHO), today, the number of people aged 60 years and older will outnumber children younger than 5 years [1]. In Brazil, "older adults" are 60 years of age or older, and it is estimated that this group will represent 18.6% in 2030 and 33.7% in 2060 of the total population [2]. The risk of cancer increases exponentially with age; about 60% of cancers occur in people of 65 years of age or older, and 70% of the deaths caused by cancers occur in this stage [3, 4].

The prevalence of malnutrition in patients with cancer varies according to the type and stage of the tumor, treatment performed, as well as age [5–7]. Chemotherapy and radiation

information about patients and their treatment, which would allow their identification. Data are available from the Ethics Committee of the Federal University of Rio Grande do Norte (contact via +55 84 9972 9763 or cep_huol@yahoo.com.br) for researchers who meet the criteria for access to confidential data.

**Funding:** The authors declare that this study received partial financial support from the Coordenação de Aperfeiçoamento de Pessoal de Nível Superior—Brasil (CAPES)—Finance Code 001. Furthermore, there was no additional external funding received for this study.

**Competing interests:** The authors have declared that no competing interests exist.

therapy cause side effects more often and in greater severity to the elderly than to the young, and frequently they can cause loss of muscle mass (MM) [8, 9]. Moreover, older cancer patients usually present loss in physical function and disability, both associated with losses of functional reserve, which, in the presence of chemotherapy, increases the likelihood that these patients will experience toxic side effects [8]. Therefore, elderly patients with cancer are easy to become sarcopenic.

The term "Sarcopenia" has originally been proposed to describe the age-related decrease in MM, today know as primary sarcopenia [10, 11]. The European Working Group on Sarcopenia in Older People (EWGSOP) suggests that sarcopenia should be evaluated through the association between reduced MM and reduced muscle function [12–14]. Since the publication of the revised version of this document (EWGSOP2), both muscle quantity and quality are accepted in the algorithm of sarcopenia [15]. Muscle quality, best described as myosteatosis, as well as the amount of MM (quantity), can be measured by the analysis of computed tomography (CT) images [16–18], a tool usually used for diagnostic of low MM in cancer patients, also known as CT-sarcopenia. However, because of occasionally unavailable technology and equipment, MM assessment remains as a problematic variable to be measured in the clinical practice of cancer patient care and in the identification of sarcopenia in these patients.

According to the revised version of guidelines of sarcopenia in older people (EWGSOP2), calf circumference (CC) has been presented as a predictor of performance and survival in older people, and may be used as a diagnostic proxy for older adults in contexts where no other MM diagnostic methods are available [15]. However, no studies comparing these two parameters of MM (CC *vs* CT image) for evaluating the prevalence of sarcopenia estimated by the EWGSOP2 reference in this population were available. Thus, the present study aimed to determine the prevalence of sarcopenia by applying the EWGSOP2 algorithm and evaluating the agreement of sarcopenia based on low CC considering sarcopenia-based CT image (MM) as a reference method in patients with cancer.

## Materials and methods

### Design and subjects

A cross-sectional study including elderly cancer patients of both genders, aged 60 years or older, in a primary care hospital, Brazil. Patients in all cancer treatment modalities (surgical, chemotherapy, radiotherapy or combined), able to perform evaluations and who had CT images of the abdominal region in the last 30 days were included. Patients with concomitant consumptive diseases (AIDS, non-cancerous liver diseases, tuberculosis), with ascites, edema or amputation, which made it impossible to analyze their CT images or measuring CC, were excluded. The study was approved by the Research Ethics Committee of the Federal University of Rio Grande do Norte (protocol number 73315617.4.0000.5292).

### Procedures

The study was conducted between January 2017 and March 2019. All eligible patients were asked about their interest in participating in the study by trained researchers at the hospital during regular consultations for cancer treatment. After verbal acceptance and signing an informed consent form, they were directed to a reserved room to assess nutritional status (anthropometry) and muscle strength. Clinical data was obtained from the digital records at the hospital and included age, sex, ethnicity, primary tumor site, treatment performed and CT images.

## Anthropometric evaluation

Three trained researchers measured body weight, height and CC. Body mass and height were determined by an electronic scale (Filizola®), with a precision of 100g. Body Mass Index (BMI) was calculated as a ratio of weight (kg) and height squared (m²), using the cut-off points proposed by the WHO: underweight ($< 18.5$ kg/m²), normal weight (18.5 to 24.9 kg/m²), overweight (25 to 29.9 kg/m²) and obese ($\geq 30$ kg/m²) [19]. For CC measurement, individuals were seated with the legs positioned at a 90˚ angle with the thigh and the inelastic band (Sanny®) around the maximum calf muscle circumference (in both legs). The measurement was performed in triplicate, and the maximum value was used. Low CC was classified using the cutoff points purposed by Barbosa-Silva et al.: 34 cm for men and 33 cm for women [20].

## Muscle strength assessment

Handgrip strength (HGS, kg) of both arms was measured using a hydraulic dynamometer (Jamar®). Patients were instructed to adjust the dynamometer and tighten it by producing as much force as possible [21]. Three attempts were made in each hand alternately, with a minimum rest period of 60 seconds for each hand [22]. The highest value recorded was used as maximum muscle strength [23]. Low HGS was determined based on the reference values of the EWGSOP2, for diagnostic of sarcopenia (HGS $< 27$kg and $< 16$kg for male and female, respectively) [15].

## Muscle mass assessment

Skeletal MM analysis was performed by evaluating using CT scans at the level of third lumbar vertebra (L3), using the Slice-O-Matic version 5.0 program (Tomovision, Montreal, Canada). A single trained researcher with anatomical knowledge selected and analyzed specific tissue using Hounsfield Unit (HU) boundaries of -29 to +150 for the skeletal muscle area (SMA, including psoas, erector spinae, lumbar square, transverse abdominal, internal and external oblique, rectus abdominis) [24]. Skeletal Muscle Index (SMI) was calculated by the total cross-sectional area (cm²) divided by height squared (m²). The SMI cut-off point proposed by Caan et al. was used to define low SMI, used as a marker of muscle quantity for the diagnostic of sarcopenia: $< 52.3$ cm²/m² for men and $< 37.6$ cm²/m² for women with a BMI $< 30$ kg/m² and $< 54.3$ cm²/m² for men and $< 46.6$ cm²/m² for women with BMI $> 30$ kg/m² [25]. Muscle quality was assessed through skeletal muscle radiodensity (SMD) from CT images and compared to the cut-off points by Kroenke et al.: $< 35.5$ HU and $< 32.5$ HU for males and females, respectively [26].

## Definition of sarcopenia

Individuals with sarcopenia were classified by two different criteria, according to the EWGSOP2; a) low HGS + reduced MM assessed by CT (including low MM quality and/or quantity, named "sarcopenia by low MM"; and b) low HGS + low CC, named "sarcopenia by low CC" [15].

## Statistical analysis

Data analysis was performed in SPSS version 22.0 for Windows. The Kolmogorov-Smirnov test was performed to assess the normality of the data. Categorical variables are expressed as absolute and relative frequency, and numerical data as mean and standard deviation. Differences in general characteristics between the sex of the patients were evaluated using the Chi-square test or Fisher's exact test for categorical variables. Differences between the quantitative

variables in patients classified with sarcopenia according to the different criteria were evaluated using independent t-test. Pearson's correlation test was performed to verify the correlation between CC and MM by CT (SMI). The Kappa coefficient between low CC and low SMI was calculated and, for its classification, the reference values considered were: $< 0.20$ as poor, 0.21–0.40 as fair, 0.41–0.60 as moderate, 0.61–0.80 as substantial, 0.81–0.99 as almost perfect, and 1.00 as perfect [27]. Sensitivity, specificity, positive and negative predictive values were calculated. A p value $< 0.05$ was considered statistically significant for all tests.

## Results

A total of 208 patients were interviewed, but 21 were unable to have their CC assessed due to leg edema or amputation. After the interview, the CT image was inaccessible for analysis in 79 patients (CT exams were older than 30 days). Thus, 108 elderly cancer patients were considered eligible, with a mean age of $70.6 \pm 7.4$ years old. Table 1 shows the clinical variables of the sample. The sex distribution is nearly even, with a slight majority of females (52.3%). Regarding the disease characteristics, the most frequent type was of colorectal cancer, followed by gastric tumor (27.8% and 22.2%, respectively), and advanced stages (III and IV) were diagnosed in more than a half of the patients (54.6%). The majority of the evaluated patients have had previous treatment (74.1%). According to BMI categories, 47.2% of patients classified with normal BMI and 30.6% with overweight. Only the variables ethnicity and cancer site showed differences between sexes.

Based on the EWGSOP2 criteria using CT data (sarcopenia by low MM), the prevalence of sarcopenia was of 24.1% (26 of 108 patients). When using the EWGSOP2 criteria with CC (sarcopenia by low CC), the prevalence was of 25.9% (28 of 108 patients). The overlap was observed in 21 individuals classified by both definitions of sarcopenia (Fig 1).

The prevalence of low CC, low HGS, and low MM (by SMI measure) were observed in 46.3%, 39.8%, and 24.1%, respectively. Table 2 presents the frequency of sarcopenia and related variables between the different criteria used. All variables presented differences between patients classified with and without sarcopenia regardless of the method used for classification. Patients with sarcopenia showed lower BMI, CC, SMA, SMI, SMD, and HGS.

The correlation between measures of CC and skeletal MM measured by CT and adjusted by height (SMI) in the sample and according to sex is presented in Fig 2. CC was weak positively correlated with SMI in the sample ($r = 0.3431$, $p < 0.001$) and in females ($r = 0.3001$, $p = 0.023$), and moderate in males ($r = 0.4573$, $p < 0.001$). Nevertheless, stronger and statistically significant correlations were observed between CC and the total MM (without dividing by height squared) analyzed by CT images. The correlations for the total sample, males and females, were $r = 0.4078$, $r = 0.6492$, and $r = 0.4611$, respectively (all p-value $<0.001$). Therefore, CC is considered a good indicator of MM. Analysis of correlation between CC and SMD were performed and showed no correlation ($r = -0.02624$, $p = 0.7875$; $r = -0.1371$, $p = 0.373$; $r = 0.01385$, $p = 0.9185$ for the general sample, males and females, respectively).

The agreement between the sarcopenia diagnostic criteria (using CT or CC for MM evaluation) is observed in Table 3. The sensitivity, specificity, and accuracy were calculated according to sex. For the total sample was observed 81% sensitivity, and 92% specificity, with higher values in females (86% sensitivity, and 91% specificity) compared to males (75% sensitivity, and 92% specificity). The Kappa tests present similar results, with a substantial agreement for females ($K = 0.729$) and males ($K = 0.673$).

## Discussion

The main finding of the present study points out that CC can be used, with good accuracy, as a MM marker to diagnose sarcopenia in elderly patients with cancer. CC is a relatively quick,

**Table 1. General characteristics of cancer patients according to sex.**

| Variables | Total (n = 108) | Male (n = 51) | Female (n = 57) | p-value |
|---|---|---|---|---|
| **Age** | | | | 0.050 |
| 60–69 years | 51 (47.2%) | 19 (37.3%) | 32 (56.1%) | |
| ≥70 years | 57 (52.8%) | 32 (62.7%) | 25 (43.9%) | |
| **Ethnicity** | | | | **0.002** |
| Caucasian | 40 (37.0%) | 11 (21.6%) | 29 (50.9%) | |
| Non-caucasian | 68 (63.0%) | 40 (78.4%) | 28 (49.1%) | |
| **Cancer site** | | | | **< 0.001** |
| Head and neck | 7 (6.5%) | 5 (9.8%) | 2 (3.5%) | |
| Gastric | 24 (22.2%) | 15 (29.4%) | 9 (15.8%) | |
| Colon and rectum | 30 (27.8%) | 14 (27.5%) | 16 (28.1%) | |
| Breast | 15 (13.9%) | - | 15 (26.3%) | |
| Prostate | 14 (13.0%) | 14 (27.5%) | - | |
| Other | 18 (16.7%) | 3 (5.9%) | 15 (26.3%) | |
| **Staging of disease** | | | | 0.078 |
| I-II | 30 (27.8%) | 9 (17.6%) | 21 (36.8%) | |
| III-IV | 59 (54.6%) | 31 (60.8%) | 28 (49.1%) | |
| Unknown | 19 (17.6%) | 11 (21.6%) | 8 (14.0%) | |
| **Previous Treatment** [1] | | | | 0.097 |
| Yes | 80 (74.1%) | 34 (66.7%) | 46 (80.7%) | |
| No | 28 (25.9%) | 17 (33.3%) | 11 (19.3%) | |
| **Nutritional status (BMI)** | | | | 0.620 |
| Underweight [2] | 9 (8.3%) | 5 (9.8%) | 4 (7.0%) | |
| Normal [3] | 51 (47.2%) | 26 (51.0%) | 25 (43.90%) | |
| Overweight [4] | 33 (30.6%) | 15 (29.4%) | 18 (31.6%) | |
| Obese [5] | 15 (13.9%) | 5 (9.8%) | 10 (17.5%) | |

Data in absolute (n) and relative (%) frequency; p value with Chi-square test and Fisher's exact test.

[1]Previous treatment including chemotherapy, radiotherapy, surgery alone or combined.

[2]BMI below 18.5 kg/m$^2$.

[3]BMI 18.5–24.9 kg/m$^2$.

[4]BMI 25.0–29.9 kg/m$^2$.

[5]BMI 30 kg/m$^2$ and above.

BMI, Body Mass Index.

cost-effective, and easy measurement that can help to identify sarcopenia without the need for sophisticated and expensive techniques. To the best of our knowledge, this is a pioneer study evaluating the accuracy of sarcopenia-based CC, compared to CT images, in this population. Previously, Velazquez-Alva et al. conducted a cross-sectional study to compare the prevalence of sarcopenia according to EWGSOP using SMI and CC in 137 Mexican elderly women [14, 28]. However, the population was composed only of women without cancer, and the diagnostic algorithm used in the study was that of the previous version published in 2010, not yet reviewed [14].

Low skeletal MM is highly prevalent in older patients with cancer and affects 5% to 89% of them depending on the type and stage of cancer [29]. The prevalence of sarcopenia in the present study is quite similar to other previous recent ones, reporting 21.2% to 48.2%, but it is important to note that the criteria to define sarcopenia used by the studies are different, being the majority only by CT images [30–34]. A recent study found a similar prevalence of

## Sarcopenia by low MM: 26          Sarcopenia by low CC: 28

Overlap sarcopenia by
low MM and by low CC
(n=21, 19.4%)

Only sarcopenia by low MM
(n=5, 4.6%)

Only sarcopenia by low CC
(n=7, 6.5%)

**Fig 1. Concordance of individual cases identified by European Working Group on Sarcopenia in Older People 2 (EWGSOP2) considering muscle mass (MM) evaluated by computed tomography (sarcopenia by low MM) and by calf circumference (CC) (sarcopenia by low CC).**

sarcopenia (27.1%) in 439 older patients with cancer (60–95 years; 43.5% women), using the same diagnostic criteria of the present study (EWGSOP2) [35]. Another important issue is that most patients of the present study had advanced cancer and had already undergone some type of treatment that can accelerate declines in MM. Williams et al., evaluating older adults before and after cancer diagnosis, observed that after cancer diagnosis, there was a decline in MM, but not in HGS or gait speed, and these declines were more striking in patients with metastases [36].

To our knowledge, few studies available in the literature have used CC as an indicator of MM in cancer patients. Our research group recently showed that low CC can predict the risk of mortality in a cohort of 250 patients with cancer [37]. Patients with low CC have a risk of death three times higher than patients with normal CC, even after adjustment for confounders; low SMI was significantly associated with mortality in crude analysis, but not after adjustment for age, sex, and stage of disease [37]. These findings reinforce the use of CC as a simple, easy,

**Table 2. Frequency of sarcopenia and comparison of quantitative variables in patients classified with sarcopenia according to the different criteria (n = 108).**

|  | Sarcopenia by low MM[1] | | | Sarcopenia by low CC[2] | | |
|---|---|---|---|---|---|---|
|  | **No** | **Yes** | **p-value** | **No** | **Yes** | **p-value** |
|  | **(82; 75.9%)** | **(26; 24.1%)** |  | **(80; 74.1%)** | **(28; 25.9%)** |  |
| Age (years) | 69.2 ± 7.6 | 74.9 ±5.2 | **<0.001** | 69.7 ± 7.7 | 73.2 ± 6.3 | **0.033** |
| Weight (Kg) | 64.1 ± 12.2 | 54.4 ± 11.9 | **0.001** | 65.8 ± 11.7 | 50.4 ± 8.2 | **<0.001** |
| BMI (kg/m$^2$) | 25.6 ± 4.4 | 22.6 ± 4.3 | **0.003** | 26.0 ± 4.3 | 21.7 ± 3.7 | **<0.001** |
| CC (cm) | 34.2 ± 3.4 | 30.6 ± 2.6 | **<0.001** | 34.6 ± 3.2 | 29.8 ± 1.9 | **<0.001** |
| SMA (cm$^2$) | 126.0 ± 30.7 | 108.3 ± 23.0 | **0.008** | 127.4 ± 30.6 | 105.8 ± 21.1 | **<0.001** |
| SMI (cm$^2$/m$^2$) | 49.9 ± 9.2 | 44.9 ± 7.9 | **0.015** | 49.9 ± 9.4 | 45.2 ± 7.1 | **<0.001** |
| SMD (HU) | 40.5 ± 8.9 | 34.2 ± 8.6 | **0.002** | 39.6 ± 9.8 | 37.3 ± 7.2 | **0.008** |
| HGS (kg/F) | 24.8 ± 9.8 | 13.8 ± 6.1 | **<0.001** | 25.3 ± 9.3 | 13.1 ± 6.4 | **<0.001** |

p-value with independent-t test comparing patients with and without sarcopenia.

[1] Low MM according to Cana et al. [25].

[2] Low CC according to Barbosa-Silva et al. [20].

BMI, Body Mass Index; CC, Calf Circumference; SMA, Skeletal Muscle Area; SMI, Skeletal Muscle Index; SMD, Skeletal Muscle Radiodensity; HU, Hounsfield Unit; HGS, Handgrip Strength.

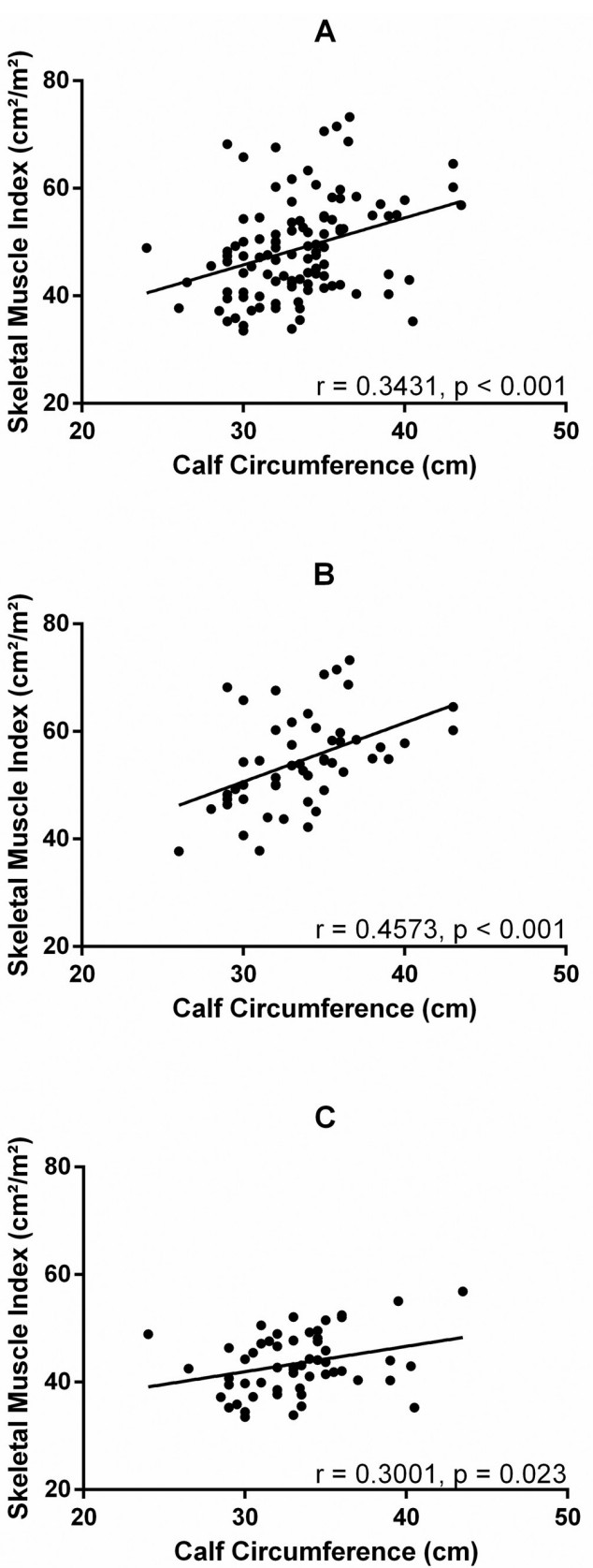

**Fig 2. Pearson's correlation between skeletal muscle index (SMI) and calf circumference in total patients (A), males (B), and females (C) with cancer (n = 108).**

cost-effective anthropometric measurement for assessing MM and screening patients with cancer.

Other studies have compared different MM indicators, including CC, for the diagnosis of sarcopenia and malnutrition in cancer patients. SARC-F is a simple and easy tool for screening sarcopenia, based on a 5-item questionnaire that measures strength, assistance in walking, rising from a chair, climbing stairs, and falls [38]. However, a major problem of SARC-F is its low sensitivity. Researches from Brazil developed an enhanced version of SARC-F (SARC-CalF) by incorporating CC into the SARC-F, which could significantly increase its sensitivity and overall diagnostic accuracy in community-dwelling older populations [39]. Because of the grown interest in this tool, other researchers also verified that SARC-CalF have better diagnostic performance as compared to the original questionnaire in the same population [40–42]. Similar results also were observed in cancer patients, when it was compared SARC-F wit SARC-CalF for screening sarcopenia in 309 advanced cancer patients [43].

Recently, the Global Leadership Initiative on Malnutrition (GLIM) criteria was proposed to identify malnutrition in adults in a clinical setting [44]. With the aim to evaluate malnutrition according to the GLIM criteria using different MM indices in lung cancer patients, Yin et al. [45] performed a multicenter, observational cohort study, and found that CC was effective for determining the nutritional status of patients, having the best performance in comparing with other anthropometric methods. In fact, other studies also showed the fair performance of CC for identifying low MM in cancer patients, and its good performance to predict mortality [37, 46, 47]. In non-cancer patients, CC was also positively correlated with skeletal MM, and it could be used as a surrogate marker of MM for diagnosing sarcopenia [20, 48].

Nowadays, there is an increase in the number of reports on body composition assessment in patients with cancer for the diagnostic of sarcopenia. The definition of sarcopenia as a state of severe depletion of skeletal MM (SMI), known as secondary sarcopenia, has been largely established for cancer patients using CT measures and defined based on the risk of mortality [49–51]. Other measures of body composition have been used in cancer patients, including dual-energy X-ray absorptiometry (DEXA) and bioelectrical impedance (BIA) [52]. Although the use of the algorithm EWGSOP2 for the diagnosis of sarcopenia in the elderly is well established for this population, researchers are not yet unanimous about the use of this reference to define sarcopenia in patients with cancer. The use of different sarcopenia criteria for cancer patients makes it difficult to compare studies, and it is suggested that this criterion should be standardized to better understand this phenomenon.

**Table 3. Accuracy test for sensitivity and specificity between different methods for low muscle mass assessment, using EWGSOP2 sarcopenia criteria in cancer patients.**

| | Total (n = 108) | Male (n = 51) | Female (n = 57) |
|---|---|---|---|
| Kappa (p-value) | 0.704 (p < 0.001) | 0.673 (p < 0.001) | 0.729 (p < 0.001) |
| Accuracy (%) | 88.9 | 88.2 | 89.5 |
| Prevalence (%) | 25.9 | 23.5 | 28.1 |
| Sensitivity (%) | 80.8 | 75.0 | 85.7 |
| Specificity (%) | 91.5 | 92.3 | 90.7 |
| Positive predictive value (%) | 75.0 | 75.0 | 75.0 |
| Negative predictive value (%) | 93.8 | 92.3 | 95.1 |

The cutoffs of low MM and low SMD varied significantly across studies, and it can represent a limitation for the present study. As we do not have any reference values developed for the Brazilian population, we used those proposed by Caan et al. [25] from a sample of American patients with stage I–III invasive colorectal cancer, and it is possible that it may not have been adequate for our population (Latin-American patients with cancer). For low CC, the cutoffs used in this study came from a regional reference, validated against DXA in a sample representative for the local population [20]. Despite the limitations, the results are relevant demonstrating the practical applicability of CC measures for the diagnostic of sarcopenia in cancer patients.

In conclusion, the agreement of sarcopenia defined by EWGSOP2 using MM assessed by CC or CT images (SMI) was moderate, with high specificity and negative predictive value, suggesting that CC can be used as a MM indicator in cancer patients. Although the application of the EWGSOP2 in the definition of sarcopenia in older cancer patients is still not a consensus, it may indeed be advantageous in clinical settings, because the measurements used in the algorithm (low HGS associated with low CC) are easier to obtain than the analysis of MM by CT images.

## Author Contributions

**Conceptualization:** Ana Paula Trussardi Fayh.

**Data curation:** Ana Paula Trussardi Fayh, Iasmin Matias de Sousa.

**Formal analysis:** Ana Paula Trussardi Fayh, Iasmin Matias de Sousa.

**Investigation:** Iasmin Matias de Sousa.

**Methodology:** Ana Paula Trussardi Fayh.

**Project administration:** Ana Paula Trussardi Fayh.

**Supervision:** Ana Paula Trussardi Fayh.

**Writing – original draft:** Ana Paula Trussardi Fayh, Iasmin Matias de Sousa.

**Writing – review & editing:** Ana Paula Trussardi Fayh, Iasmin Matias de Sousa.

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
