## [Decision Letter · Decision Letter 0]

15 Feb 2021

PONE-D-20-33755

Comparison of revised EWGSOP2 criteria of sarcopenia in patients with cancer using different parameters of muscle mass

PLOS ONE

Dear Dr. Trussardi Fayh,

Thank you for submitting your manuscript to PLOS ONE. After careful consideration, we feel that it has merit but does not fully meet PLOS ONE’s publication criteria as it currently stands.  Therefore, we invite you to submit a revised version of the manuscript that addresses the points raised during the review process by the expert Reviewers.

We look forward to receiving your revised manuscript.

Kind regards,

Stephen E Alway, Ph.D.

Academic Editor

PLOS ONE

Journal Requirements:

"This study was financed in part by the Coordenação de Aperfeiçoamento de

306 Pessoal de Nível Superior - Brasil (CAPES) - Finance Code 001."

Reviewers' comments:

Reviewer's Responses to Questions

**Comments to the Author**

1. Is the manuscript technically sound, and do the data support the conclusions?

Reviewer #1: Yes

Reviewer #2: Yes

2. Has the statistical analysis been performed appropriately and rigorously? 

Reviewer #1: Yes

Reviewer #2: Yes

3. Have the authors made all data underlying the findings in their manuscript fully available?

Reviewer #1: Yes

Reviewer #2: Yes

4. Is the manuscript presented in an intelligible fashion and written in standard English?

Reviewer #1: Yes

Reviewer #2: Yes

5. Review Comments to the Author

Reviewer #1: ‘Comparison of revised EWGSOP2 criteria of sarcopenia in patients with cancer using different parameters of muscle mass’ is a research article that is a pioneer study evaluating the accuracy of sarcopenia-based CC, compared to CT images.

This study is very valuable because it highlights the fact that calf circumference measurement can be used to muscle mass assessment for sarcopenia diagnosis in cancer patients, instead of computed tomography (CT). Calf circumference measurement is a simple, easy and cheap procedure, in opposite to CT, which is costly method and requiring specialized staff to operate it. The paper reads well.

I would only recommend to widen the description of the use of the SARC-CalF questionnaire in research on sarcopenia around the world, because interest in this tool has grown significantly in recent years (lines 256-258).

Please refer to the following articles:

1) Yang M, Hu X, Xie L, et al. Screening sarcopenia in community-dwelling older adults: SARC-F vs SARC-F combined with calf circumference (SARC-CalF). J Am Med Dir Assoc. 2018;19(3):277.e1-277.e8. doi:10.1016/j.jamda.2017.12.016

2) Krzymińska-Siemaszko R, Deskur-Śmielecka E, Kaluźniak-Szymanowska A, Lewandowicz M, Wieczorowska-Tobis K. Comparison of Diagnostic Performance of SARC-F and Its Two Modified Versions (SARC-CalF and SARC-F+EBM) in Community-Dwelling Older Adults from Poland. Clin Interv Aging. 2020; 15: 583–594.

3) Mo Y, Dong X, Wang XH. Screening accuracy of SARC-F combined with calf circumference for sarcopenia in older adults: a diagnostic meta-analysis. J Am Med Dir Assoc. 2020;21(2):288–289. doi:10.1016/j.jamda.2019.09.002

Reviewer #2: This manuscript reports the comparison of sarcopenia in cancer patients using computed tomography and calf circumference. Overall the study is important and it will certainly add to the current evidence on the potential of using a simpler method to measure muscle mass in cancer patients. However, I have some minor comments on the manuscript, which are outlined below,

Abstract

• Please write L3 in full when it's first mentioned in the text, i.e. "third lumbar vertebra".

• Mean ± “standard deviation” age of 70.6 ± 7.4 years?

Materials and Methods

• Was this trial registered in clinical trials registration? If so, please include this information in the Methods section.

Design and subjects

• Lines 83-84: Please include the institution of the ethics board.

Anthropometric evaluation

• Lines 96-99: How many measurements were taken for the calf circumference, e.g. once or twice?

Muscle strength assessment

• Lines 104-105: Is there any reason why the handgrip strength of the non-dominant hand was captured as well?

Results

• Table 1: Please include the definition for each of the BMI categories.

• Line 161: Should this read "Chi-square test"?

• Line 180: Please write the abbreviations in full when first mentioned.

• Table 2: It will be good to include the prevalence of low handgrip strength, low MM, and low CC in this table. Thank you.

• Line 183: Please change “according” to "according to".

• Table 2 header: Please include the denominator for all the "No" and "Yes" categories, e.g. Sarcopenia by low MM, Yes: "26 of 108 patients = 24.1%".

• Line 194: Should this read "moderate"?

• Table 3: Are all the kappa p-values referring to "p<0.001"?

Discussion

• Lines 250-251: Suggest using another term for 'patients at risk of death'.

• Line 256: Should this read 'researchers"?

6. PLOS authors have the option to publish the peer review history of their article (what does this mean?). If published, this will include your full peer review and any attached files.

Reviewer #1: No

Reviewer #2: No

---

## [Author Response · Author response to Decision Letter 0]

15 Mar 2021

February 17th, 2021.

RESPONSE LETTER

Manuscript PONE-D-20-33755

Comparison of revised EWGSOP2 criteria of sarcopenia in patients with cancer using different parameters of muscle mass

Dear Stephen E Alway, Ph.D.

Academic Editor:

Thank you for your email enclosing the reviewers’ comments. We have carefully reviewed the comments and have revised the manuscript accordingly. Our responses are given point-by-point below. Changes to the manuscript are tracked.

Sincerely,

Ana Paula Trussardi Fayh, PhD

Editor comments: 

Response: The manuscript meets the style and templates provided by the journal. 

"This study was financed in part by the Coordenação de Aperfeiçoamento de

306 Pessoal de Nível Superior - Brasil (CAPES) - Finance Code 001."

Response: Thank you for the comment. The statement was included in the submission form and removed from the manuscript. The cover letter was also changed accordingly.

Reviewers' comments:

 Reviewer #1: 

‘Comparison of revised EWGSOP2 criteria of sarcopenia in patients with cancer using different parameters of muscle mass’ is a research article that is a pioneer study evaluating the accuracy of sarcopenia-based CC, compared to CT images.

This study is very valuable because it highlights the fact that calf circumference measurement can be used to muscle mass assessment for sarcopenia diagnosis in cancer patients, instead of computed tomography (CT). Calf circumference measurement is a simple, easy and cheap procedure, in opposite to CT, which is costly method and requiring specialized staff to operate it. The paper reads well.

I would only recommend to widen the description of the use of the SARC-CalF questionnaire in research on sarcopenia around the world, because interest in this tool has grown significantly in recent years (lines 256-258).

Please refer to the following articles:

1) Yang M, Hu X, Xie L, et al. Screening sarcopenia in community-dwelling older adults: SARC-F vs SARC-F combined with calf circumference (SARC-CalF). J Am Med Dir Assoc. 2018;19(3):277.e1-277.e8. doi:10.1016/j.jamda.2017.12.016

2) Krzymińska-Siemaszko R, Deskur-Śmielecka E, Kaluźniak-Szymanowska A, Lewandowicz M, Wieczorowska-Tobis K. Comparison of Diagnostic Performance of SARC-F and Its Two Modified Versions (SARC-CalF and SARC-F+EBM) in Community-Dwelling Older Adults from Poland. Clin Interv Aging. 2020; 15: 583–594.

3) Mo Y, Dong X, Wang XH. Screening accuracy of SARC-F combined with calf circumference for sarcopenia in older adults: a diagnostic meta-analysis. J Am Med Dir Assoc. 2020;21(2):288–289. doi:10.1016/j.jamda.2019.09.002

Response: Thank you for the recommendation. Additional informations were included in the manuscript to refer to these studies.

Reviewer #2: Reviewer #2: This manuscript reports the comparison of sarcopenia in cancer patients using computed tomography and calf circumference. Overall the study is important and it will certainly add to the current evidence on the potential of using a simpler method to measure muscle mass in cancer patients. However, I have some minor comments on the manuscript, which are outlined below,

Abstract

• Please write L3 in full when it's first mentioned in the text, i.e. "third lumbar vertebra".

Response: Thank you for observation.

• Mean ± “standard deviation” age of 70.6 ± 7.4 years?

Response: Thank you for your observation, it was corrected according to the suggestion.

Materials and Methods

• Was this trial registered in clinical trials registration? If so, please include this information in the Methods section.

Response: Thank you for your observation. Considering the fact that the design of the study is cross-sectional (observational), no clinical trial registration was performed.

Design and subjects

• Lines 83-84: Please include the institution of the ethics board.

Response: Thank you for your observation. The institution was included according to the suggestion.

Anthropometric evaluation

• Lines 96-99: How many measurements were taken for the calf circumference, e.g. once or twice?

Response: Thank you for your observation. The information about the measurements was corrected and included in the manuscript (not in the dominant leg; both leg calf muscle circumferences were measured).

Muscle strength assessment

• Lines 104-105: Is there any reason why the handgrip strength of the non-dominant hand was captured as well?

Response: Thank you for your valuable observation. There is a mistake in this information, because handgrip strength was measured in both arms.

Results

• Table 1: Please include the definition for each of the BMI categories.

Response: Thank you for your observation. The definition was included in the footnote Table.

• Line 161: Should this read "Chi-square test"?

Response: Thank you for your observation, the term was modified.

• Line 180: Please write the abbreviations in full when first mentioned.

Response: Thank you for your observation. The abbreviations were defined in following sentences: BMI (line 97), CC (line 68), SMA (line 119), SMI (line 121), SMD (line 127) and HGS (line 107)

• Table 2: It will be good to include the prevalence of low handgrip strength, low MM, and low CC in this table. Thank you.

Response: Thank you for the recommendation to include these prevalences. However, we decided to present this information in the text (line 181-182). 

• Line 183: Please change “according” to "according to".

Response: Thank you for your observation, the term was changed.

• Table 2 header: Please include the denominator for all the "No" and "Yes" categories, e.g. Sarcopenia by low MM, Yes: "26 of 108 patients = 24.1%".

Response: Thank you for your observation, the number of subjects was added on the title of Table.

• Line 194: Should this read "moderate"?

Response: Thank you for your observation, we corrected the word in the sentence.

• Table 3: Are all the kappa p-values referring to "p<0.001"?

Response: Thank you for your observation. Yes, all kappa p-values are p < 0.001. We changed the manuscript accordingly.

Discussion

• Lines 250-251: Suggest using another term for 'patients at risk of death'.

Response: Thank you for your observation, we changed the term in the sentence limiting to the studied population. 

• Line 256: Should this read 'researchers"?

Response: Thank you for your observation, we corrected the word in the sentence.

---

## [Decision Letter · Decision Letter 1]

21 Jun 2021

PONE-D-20-33755R1

Comparison of revised EWGSOP2 criteria of sarcopenia in patients with cancer using different parameters of muscle mass

PLOS ONE

Dear Dr. Trussardi Fayh,

Thank you for submitting your manuscript to PLOS ONE. After careful consideration, we feel that it has merit but does not fully meet PLOS ONE’s publication criteria as it currently stands. Therefore, we invite you to submit a revised version of the manuscript that addresses the points raised during the review process.

This paper has only a few minor things to address. All of the major issues have been resolved so i am confident that the next revision can be done quickly.

We look forward to receiving your revised manuscript.

Kind regards,

Stephen E Alway, Ph.D.

Academic Editor

PLOS ONE

Journal Requirements:

Reviewers' comments:

Reviewer's Responses to Questions

**Comments to the Author**

1. If the authors have adequately addressed your comments raised in a previous round of review and you feel that this manuscript is now acceptable for publication, you may indicate that here to bypass the “Comments to the Author” section, enter your conflict of interest statement in the “Confidential to Editor” section, and submit your "Accept" recommendation.

Reviewer #1: All comments have been addressed

Reviewer #3: All comments have been addressed

2. Is the manuscript technically sound, and do the data support the conclusions?

Reviewer #1: Yes

Reviewer #3: Yes

3. Has the statistical analysis been performed appropriately and rigorously? 

Reviewer #1: Yes

Reviewer #3: I Don't Know

4. Have the authors made all data underlying the findings in their manuscript fully available?

Reviewer #1: Yes

Reviewer #3: No

5. Is the manuscript presented in an intelligible fashion and written in standard English?

Reviewer #1: Yes

Reviewer #3: Yes

6. Review Comments to the Author

Reviewer #1: Thank you for considering my comments. I recommend publishing this revised version of manuscript titled Comparison of revised EWGSOP2 criteria of sarcopenia in patients with cancer using different parameters of muscle mass.

Reviewer #3: Overall, I think the authors have done a good job at addressing the previous round of revisions. Below are a few thoughts and minor suggestions for improvement.

General Comments:

Double check all abbreviations are used consistently in paper. E.g. “Muscle mass” has frequently been abbreviated to MM. However, in some cases it still remains written in full.

Abstract:

Please introduce European 56 Work Group on Sarcopenia in Older People (EWGSOP) in abstract before referring to abbreviation.

Line with “prevalence of sarcopenia by the EWGSOP2” – suggest revising to “according to the EWGSOP2 criteria.”

Line with “evaluate the accuracy of sarcopenia using low CC considering MM assessment by computed tomography (CT)…” – suggest revising to “evaluate the accuracy of sarcopenia using low CC relative to MM assessment…”

Line “Data included sociodemographic, clinical and anthropometric variables.” – suggest deleting from abstract if you need to reduce word count. I don’t think this is necessary.

Suggest adding one additional statement at the end of the abstract or slightly revise the current last statement of the abstract. What is the key takeaway message from this study, i.e. what is the primary reason people will cite this paper? E.g. Measuring CC and HGS may serve as a useful alternative to identify sarcopenia in people with cancer relative to relying on CT imaging to diagnose low MM.

Introduction:

Line 60: remove capitalisation of sarcopenia.

Methods:

Line 90: This needs further clarification. How were patients approached for the study? Was there a set time period for recruitment, e.g. mm-yy to mm-yy?

Line 101: Was the calf measurement performed by the same researcher?

Line 103: After three measures of CC were taken, was the average then calculated across both legs or was it the maximum value that was used? This needs to be clearly articulated to allow for study replication or translation into practice.

Results:

Line 152: Was there a primary reason patients were unable to have their calf assessed? A little further detail might be helpful here.

Line 160: I would not refer to this as a nutritional evaluation, as that would require a more comprehensive assessment beyond BMI alone. Suggest revising this to: According to BMI categories, 47.2% of patients were classified as having a normal body weight and 30.6% were considered overweight.”

Line 202-207: This is a very long sentence. Suggest condensing or breaking into two or more statements for clarity. Please also double check use of brackets. One seems to be missing before “all p-value <0.001.”

Discussion:

Line 228: This sentence is a bit difficult to follow. Try to be very clear here. E.g. “The main finding of the present study is that CC can be used to diagnose sarcopenia in elderly patients with cancer and has similar accuracy to assessing MM using CT imaging.”

Line 230: Remove “In fact”

The utility of measuring CC versus relying on sophisticated and costly imaging techniques to identify low MM should be highlighted. One of the most important findings of this study is that a relatively quick, cost-effective and easy measurement can help identify sarcopenia. I suggest highlighting this more in the discussion.

7. PLOS authors have the option to publish the peer review history of their article (what does this mean?). If published, this will include your full peer review and any attached files.

Reviewer #1: **Yes: **Roma Krzymińska-Siemaszko

Reviewer #3: No

---

## [Author Response · Author response to Decision Letter 1]

22 Jun 2021

June 22nd, 2021.

RESPONSE LETTER

Manuscript PONE-D-20-33755R1

Comparison of revised EWGSOP2 criteria of sarcopenia in patients with cancer using different parameters of muscle mass

Dear Stephen E Alway, Ph.D.

Academic Editor:

Thank you for your email enclosing the reviewers’ comments. We have carefully reviewed the comments and have revised the manuscript accordingly. Our responses are given point-by-point below. Changes to the manuscript are tracked.

Sincerely,

Ana Paula Trussardi Fayh, PhD

Reviewers' comments:

Reviewer #1

Thank you for considering my comments. I recommend publishing this revised version of manuscript titled Comparison of revised EWGSOP2 criteria of sarcopenia in patients with cancer using different parameters of muscle mass.

Response: Thank you for your comments; we appreciate your contributions to our manuscript.

Reviewer #3

Double check all abbreviations are used consistently in paper. E.g. “Muscle mass” has frequently been abbreviated to MM. However, in some cases it still remains written in full.

Response: Thank you for your observation; we have double-checked the main document and now all mentions of muscle mass after the abbreviation are abbreviated in the manuscript.

Abstract:

- Please introduce European Work Group on Sarcopenia in Older People (EWGSOP) in abstract before referring to abbreviation.

Response: Thank you for point this out. We have included the meaning of the abbreviation in the first mention. 

- Line with “prevalence of sarcopenia by the EWGSOP2” – suggest revising to “according to the EWGSOP2 criteria.”

Response: Thank you for your comment, we have adjusted the sentence accordingly.

- Line with “evaluate the accuracy of sarcopenia using low CC considering MM assessment by computed tomography (CT)…” – suggest revising to “evaluate the accuracy of sarcopenia using low CC relative to MM assessment…”

Response: Thank you for your observation. We have changed the main document. 

- Line “Data included sociodemographic, clinical and anthropometric variables.” – suggest deleting from abstract if you need to reduce word count. I don’t think this is necessary.

Response: Thank you for your observation. However, there was no need to reduce the word count after the changes in the abstract, the word count remains according to the journal guidelines (not exceed 300 words).

- Suggest adding one additional statement at the end of the abstract or slightly revise the current last statement of the abstract. What is the key takeaway message from this study, i.e. what is the primary reason people will cite this paper? E.g. Measuring CC and HGS may serve as a useful alternative to identify sarcopenia in people with cancer relative to relying on CT imaging to diagnose low MM.

Response: Thank you for your suggestion. We included additional information in the conclusion of the abstract to support the takeaway message of our study. 

Introduction:

-Line 60: remove capitalisation of sarcopenia.

Response: Thank you for your observation. The capitalization was removed.

Methods:

- Line 90: This needs further clarification. How were patients approached for the study? Was there a set time period for recruitment, e.g. mm-yy to mm-yy?

Response: Thank you for your comments. We have included this information in the manuscript (lines 94 to 96).

- Line 101: Was the calf measurement performed by the same researcher?

Response: Thank you for your observation. Three trained researchers performed the calf measurement. This information was added in the revised manuscript (line 102).

- Line 103: After three measures of CC were taken, was the average then calculated across both legs or was it the maximum value that was used? This needs to be clearly articulated to allow for study replication or translation into practice.

Response: Thank you for your comments. After the three measures, it was used the maximum value. The additional information was included in the manuscript (line 110).

Results:

- Line 152: Was there a primary reason patients were unable to have their calf assessed? A little further detail might be helpful here.

Response: Thank you for your observation. The patients were unable to have their calf assessed because of the presence of leg edema or amputation. Additional information is provided in the revised version of the manuscript (Methods section, line 89 and Results section, line 159).

- Line 160: I would not refer to this as a nutritional evaluation, as that would require a more comprehensive assessment beyond BMI alone. Suggest revising this to: According to BMI categories, 47.2% of patients were classified as having a normal body weight and 30.6% were considered overweight.”

Response: Thank you for your suggestion. We have agreed with your comment and changed the manuscript according to your recommendation.

- Line 202-207: This is a very long sentence. Suggest condensing or breaking into two or more statements for clarity. Please also double check use of brackets. One seems to be missing before “all p-value <0.001.”

Response: Thank you for your observation. We have revised the sentence in the manuscript.

Discussion:

- Line 228: This sentence is a bit difficult to follow. Try to be very clear here. E.g. “The main finding of the present study is that CC can be used to diagnose sarcopenia in elderly patients with cancer and has similar accuracy to assessing MM using CT imaging.”

Response: Thank you for your comment. However, the sentence in the document is “The main finding of the present study points out that CC can be used, with good accuracy, as a MM marker to diagnose sarcopenia in elderly patients with cancer.”

-Line 230: Remove “In fact”

Response: Thank you for your observation. We have removed the sentence in the revised manuscript.

- The utility of measuring CC versus relying on sophisticated and costly imaging techniques to identify low MM should be highlighted. One of the most important findings of this study is that a relatively quick, cost-effective and easy measurement can help identify sarcopenia. I suggest highlighting this more in the discussion.

Response: Thank you for your suggestion. We agreed and included a sentence in the discussion section to highlight the importance of the use of CC (lines 237 to 238).

---

## [Decision Letter · Decision Letter 2]

2 Sep 2021

Comparison of revised EWGSOP2 criteria of sarcopenia in patients with cancer using different parameters of muscle mass

PONE-D-20-33755R2

Dear Dr. Trussardi Fayh,

The expert Reviewers who reviewed your manuscript are satisfied that the concerns have been addressed in this revision and I agree. Therefore we are pleased to inform you that your manuscript has been judged scientifically suitable for publication and will be formally accepted for publication once it meets all outstanding technical requirements.

Congratulations on a great paper, and kind regards,

Stephen E Alway, Ph.D.

Academic Editor

PLOS ONE

Additional Editor Comments (optional):

Reviewers' comments:

Reviewer's Responses to Questions

**Comments to the Author**

1. If the authors have adequately addressed your comments raised in a previous round of review and you feel that this manuscript is now acceptable for publication, you may indicate that here to bypass the “Comments to the Author” section, enter your conflict of interest statement in the “Confidential to Editor” section, and submit your "Accept" recommendation.

Reviewer #3: All comments have been addressed

2. Is the manuscript technically sound, and do the data support the conclusions?

Reviewer #3: Yes

3. Has the statistical analysis been performed appropriately and rigorously? 

Reviewer #3: Yes

4. Have the authors made all data underlying the findings in their manuscript fully available?

Reviewer #3: Yes

5. Is the manuscript presented in an intelligible fashion and written in standard English?

Reviewer #3: Yes

6. Review Comments to the Author

Reviewer #3: The authors have carefully gone through the suggested comments and revised the manuscript accordingly. I feel the manuscript is improved and have no further comments.

7. PLOS authors have the option to publish the peer review history of their article (what does this mean?). If published, this will include your full peer review and any attached files.

Reviewer #3: No

---

## [Editor Report · Acceptance letter]

6 Sep 2021

PONE-D-20-33755R2 

Comparison of revised EWGSOP2 criteria of sarcopenia in patients with cancer using different parameters of muscle mass 

Dear Dr. Trussardi Fayh:

I'm pleased to inform you that your manuscript has been deemed suitable for publication in PLOS ONE. Congratulations! Your manuscript is now with our production department. 

Kind regards, 

on behalf of

Dr. Stephen E Alway 

Academic Editor

PLOS ONE